# Rapid emergence of extensively drug-resistant *Shigella sonnei* in France

Sophie Lefèvre [1], Elisabeth Njamkepo[1], Sarah Feldman [2,3], Corinne Ruckly[1], Isabelle Carle[1], Monique Lejay-Collin[1], Laëtitia Fabre[1], Iman Yassine [1], Lise Frézal[1], Maria Pardos de la Gandara [1], Arnaud Fontanet[2] & François-Xavier Weill [1] ✉

*Shigella sonnei*, the main cause of bacillary dysentery in high-income countries, has become increasingly resistant to antibiotics. We monitored the antimicrobial susceptibility of 7121 *S. sonnei* isolates collected in France between 2005 and 2021. We detected a dramatic increase in the proportion of isolates simultaneously resistant to ciprofloxacin (CIP), third-generation cephalosporins (3GCs) and azithromycin (AZM) from 2015. Our genomic analysis of 164 such extensively drug-resistant (XDR) isolates identified 13 different clusters within CIP-resistant sublineage 3.6.1, which was selected in South Asia ~15 years ago. AZM resistance was subsequently acquired, principally through IncFII (pKSR100-like) plasmids. The last step in the development of the XDR phenotype involved various extended-spectrum beta-lactamase genes (*bla*CTX-M-3, *bla*CTX-M-15, *bla*CTX-M-27, *bla*CTX-M-55, and *bla*CTX-M-134) carried by different plasmids (IncFII, IncI1, IncB/O/K/Z) or even integrated into the chromosome, and encoding resistance to 3GCs. This rapid emergence of XDR *S. sonnei*, including an international epidemic strain, is alarming, and good laboratory-based surveillance of shigellosis will be crucial for informed decision-making and appropriate public health action.

*Shigella* species (now serogroups) are specialized lineages of *Escherichia coli* causing invasive intestinal infections in humans, ranging from acute watery diarrhea to dysenteric syndrome. Most people recover spontaneously from shigellosis, but antibiotic therapy is recommended for adults and children with bloody diarrhea, patients at risk of complications or to stop transmission in certain outbreak-prone settings[1–3]. The drugs currently used are ciprofloxacin (CIP), ceftriaxone (a third-generation cephalosporin, 3GC), and azithromycin (AZM). There are four serogroups of *Shigella*, with *S. sonnei* the predominant serogroup circulating in industrialized countries and emerging worldwide, even in countries in which other *Shigella* serogroups have traditionally predominated[4]. In 2019, 8848 confirmed cases of

shigellosis were reported by 30 countries of the European Union (including the UK) and the European Economic Area; 59.4% were due to *S. sonnei*[5]. Many multidrug-resistant (MDR) strains of *S. sonnei* have been described in recent years. MDR strains were originally linked to travel to Asia or circulation in particular communities or networks, such as gay, bisexual and other men who have sex with men (GBMSM)[6–14]. However, *S. sonnei* isolates simultaneously resistant to the three recommended antimicrobial drugs (CIP, 3GCs, and AZM), described as "extensively drug-resistant" (XDR), were reported only exceptionally before 2022 and were often linked to Southeast Asia[7,9,11]. A study in the context of high-income countries revealed that only 0.8% (36/4222) of the 4222 *S. sonnei* genomic sequences obtained in

[1]Institut Pasteur, Université Paris Cité, Unité des Bactéries pathogènes entériques, Centre National de Référence des Escherichia coli, Shigella et Salmonella, Paris F-75015, France. [2]Institut Pasteur, Université Paris Cité, Unité Epidémiologie des maladies émergentes, Paris F-75015, France. [3]Present address: National Institute for Antibiotic Resistance and Infection Control, Ministry of Health, Tel-Aviv Sourasky Medical Center, Tel Aviv 6423906, Israel. ✉e-mail: francois-xavier.weill@pasteur.fr

the framework of public health surveillance in England, Australia and the US between 2016 and 2019 were inferred to be XDR[15]. However, no phenotypic antimicrobial susceptibility data were provided to confirm the XDR phenotype. In 2022, the United Kingdom Health Security Agency (UKHSA) and the European Center for Disease Prevention and Control (ECDC) reported an increase in XDR *S. sonnei* infections in 10 European countries, with France and the UK the most seriously affected[5].

In this study, based on a combination of long-term conventional laboratory surveillance data and high-resolution whole-genome analyses, we investigated the timeline of XDR *S. sonnei* emergence in France, and the genomic diversity and recent evolution of these strains.

## Results and discussion
### Antimicrobial susceptibility data of *S. sonnei* in France
Our review of *S. sonnei* antimicrobial susceptibility data obtained between 2005 and 2021 (based on 7121 isolates received and confirmed at the French National Reference Center for *E. coli*, *Shigella* and *Salmonella*, FNRC-ESS, Institut Pasteur) revealed a sharp increase in the percentage of isolates resistant to 3GCs, CIP and AZM ($P < 0.001$) (Fig. 1). A first isolate resistant to 3GCs was identified as early as 2005 (prevalence of 1/342, 0.3%), whereas the first isolates resistant to CIP were identified in 2008 (prevalence of 2/373, 0.5%). Ten isolates resistant to AZM were detected in 2014, the year in which screening for such resistance began (in April) (10/557, 1.8%). In 2021, 29.5% (131/444) of *S. sonnei* isolates were resistant to 3GCs, 42.3% (188/444) were resistant to CIP and 38.7% (172/444) were resistant to AZM (Fig. 1). The first XDR *S. sonnei* isolate was identified in 2015, and an additional 163 XDR isolates have since been obtained (Fig. 1 and Supplementary Tables 1 and 2). All but one of these 164 XDR isolates were collected from mainland France. The percentage of XDR *S. sonnei* isolates increased over the study period ($P < 0.001$), peaking at 22.3% (99/444) in 2021. All but one of these XDR isolates were also resistant to trimethoprim/sulfamethoxazole and 87.2% (143/164) were resistant to tetracyclines. All these isolates remained susceptible to carbapenems.

One limitation to this study is that shigellosis is not a mandatory notifiable disease in France. Thus, laboratory-based surveillance, despite its official introduction 50 years ago, captures only a portion of the shigellosis burden (see Methods). The COVID-19 pandemic also

had a clear impact on our surveillance data, as the number of *S. sonnei* infections in France decreased sharply in 2020 (as also reported elsewhere in Europe[5]) and, to a lesser extent, in 2021. This decrease may reflect a true decrease in the number of cases due to travel restrictions (43 and 30% decreases in the proportion of *S. sonnei* cases reporting international travel in 2020 and 2021, respectively, relative to 2019, in our surveillance data), stricter hygiene measures, lockdowns and school closures. However, it may also reflect poorer access to the healthcare system for patients with mild infections. In addition, it is not possible to rule out better referral to the FNRC-ESS of unusual XDR isolates by the clinical laboratories participating in our network. The high percentage of XDR *S. sonnei* observed in 2021 might, therefore, constitute an overestimate. However, in 2022, the year in which most public health measures against COVID-19 were lifted in France, XDR *S. sonnei* still accounted for 21.7% (73/336) of all *S. sonnei* isolates received from January 1 to August 31 at the FNRC-ESS.

### Phylogenomics of the XDR *S. sonnei* isolates in France
We investigated phylogenetic relationships by sequencing the genomes of the 164 French XDR *S. sonnei* isolates. We also included 2976 genomes from isolates and historical strains from the FNRC-ESS collected between 1945 and 2021 in the phylogenomic analysis, to provide a phylogenetic context for the XDR *S. sonnei* isolates. A maximum likelihood (ML) phylogenetic tree was built from an alignment of 59,295 chromosomal recombination-filtered single-nucleotide variants (SNVs) to obtain an accurate view of the evolutionary relationships (vertical evolution) between *S. sonnei* strains. This ML tree revealed that the 164 XDR *S. sonnei* isolates were not grouped into a single cluster, instead forming 13 different clusters (X1 to X13), all within lineage L3, the predominant lineage of *S. sonnei* worldwide over the last two decades (Fig. 2)[15,16].

Using a new genomic tool based on a selection of core-genome SNVs developed by Hawkey et al. [15] for categorizing *S. sonnei* into informative phylogenetic genotypes with a universal nomenclature, we assigned the 164 XDR isolates to four different subtypes of subclade 3.6.1 (Fig. 2). The two clusters with the largest numbers of XDR isolates were X10 ($n = 102$) belonging to genotype 3.6.1.1.2_CipR.MSM5 (lineage L3, clade 6, subclade 1, subtype 1.2 with alias name CipR.MSM5, as used in previous publications for isolates resistant to

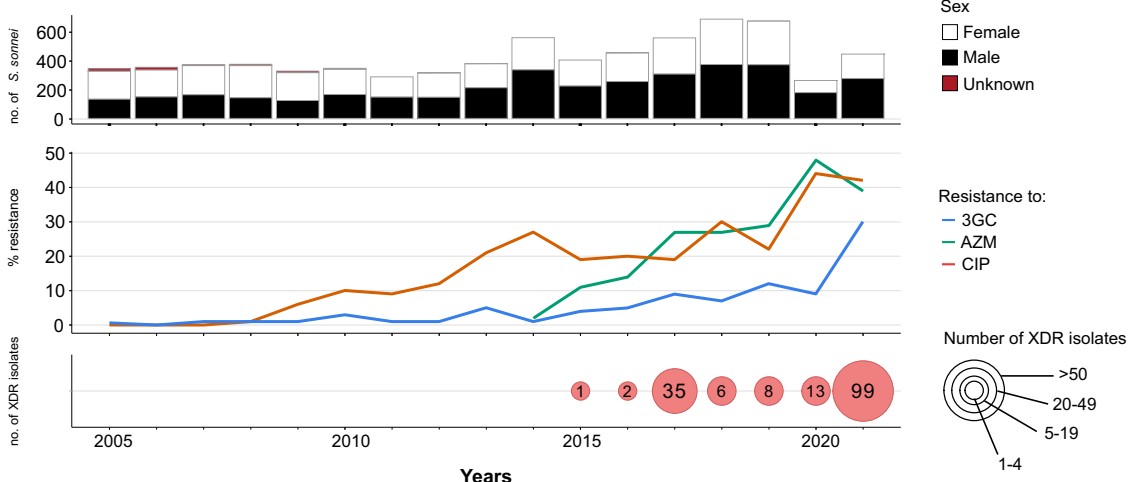

**Fig. 1 | Laboratory surveillance of *Shigella sonnei* infections in France, 2005–2021.** The upper panel shows the number of isolates, by patient sex, received and analyzed per year. The middle panel shows the percentage of isolates displaying phenotypic resistance to ciprofloxacin (CIP), azithromycin (AZM), or third-generation cephalosporins (3GCs). The lower panel shows the number of extensively drug-resistant (XDR) *S. sonnei* isolates detected per year. XDR means combined resistance to CIP, AZM, and 3GCs.

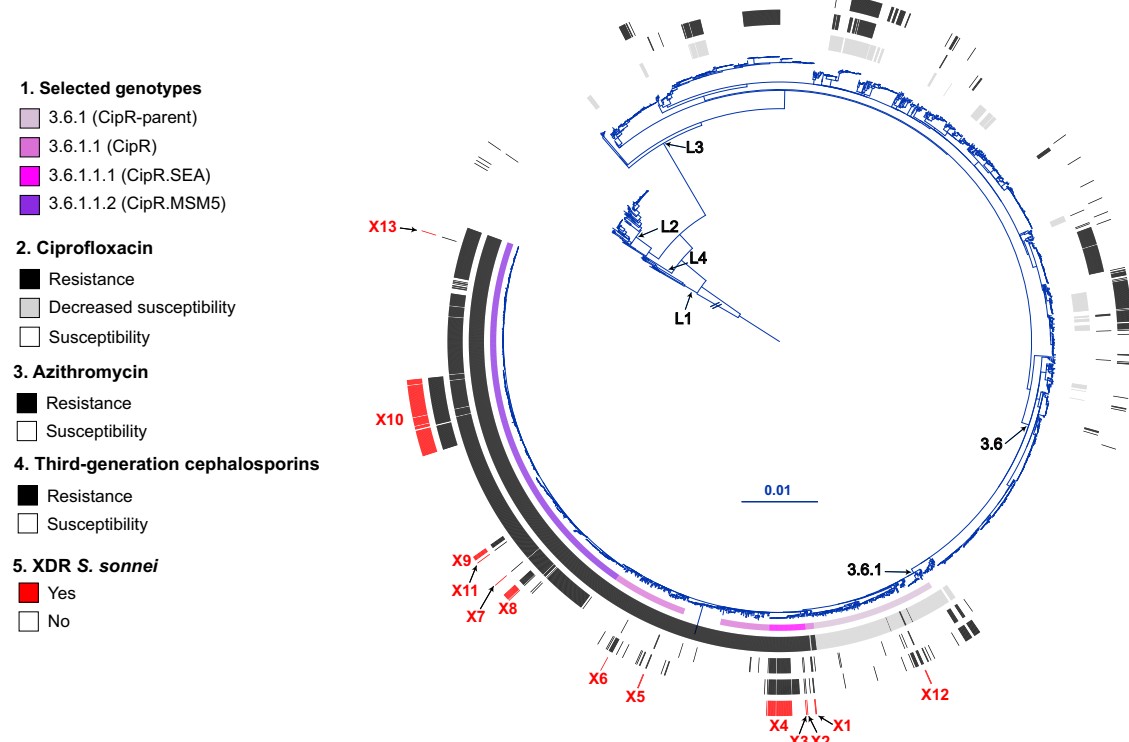

**1. Selected genotypes**
- 3.6.1 (CipR-parent)
- 3.6.1.1 (CipR)
- 3.6.1.1.1 (CipR.SEA)
- 3.6.1.1.2 (CipR.MSM5)

**2. Ciprofloxacin**
- Resistance
- Decreased susceptibility
- Susceptibility

**3. Azithromycin**
- Resistance
- Susceptibility

**4. Third-generation cephalosporins**
- Resistance
- Susceptibility

**5. XDR *S. sonnei***
- Yes
- No

**Fig. 2 | Maximum-likelihood phylogeny of 3141 *S. sonnei* genomic sequences.** The *S. sonnei* reference genome 53G (GenBank accession numbers NC_016822) was also included in addition to the 3140 genomes from the French National Reference Center for *E. coli*, *Shigella,* and *Salmonella* (Supplementary Data 1). The circular phylogenetic tree (in blue) is rooted on *S. flexneri* 2a strain 2457T and its branch length has been shortened by a factor of 100 (indicated by the double slash) to improve vizualisation. *S. sonnei* lineages L1 to L4, clade 3.6, and subclade 3.6.1 are indicated. The rings show the associated information (see key) for each isolate, according to its position in the phylogeny, from the innermost to the outermost, in the following order: (1) selected *S. sonnei* genotypes; (2) antimicrobial susceptibility testing (AST) for ciprofloxacin; (3) AST for azithromycin; (4) AST for third-generation cephalosporins; (5) and the XDR isolates (with the XDR cluster names, X1 to X13, in red). The scale bar indicates the number of substitutions per variable site (SNVs). Due to a deliberate strategy to enrich our genomic dataset with azithromycin-resistant, ciprofloxacin-resistant, and third-generation cephalosporin-resistant isolates collected before 2017, the year in which routine genomic surveillance began in France (see "Methods"), there is an overrepresentation of these resistances in this figure with respect to the global population of 7121 isolates studied.

CIP in GBMSM[10,15]), and X4 (*n* = 34) belonging to genotype 3.6.1.1.1_CipR.SEA (lineage L3, clade 6, subclade 1, subtype 1.1 with alias name CipR.SEA, as used in previous publications for isolates resistant to CIP in Southeast Asia[6,15]) (Fig. 3). Some of the XDR *S. sonnei* isolates belonging to genotypes 3.6.1_CipR-parent, 3.6.1.1_CipR and 3.6.1.1.1_CipR.SEA were acquired following travel (mostly to South and Southeast Asia) (Fig. 3). For cluster X4 (genotype 3.6.1.1.1_CipR.SEA), three patients reported travel to Southeast Asia, including one who was the index case of an outbreak at an elementary school (91 cases identified) in 2017, leading to the temporary closure of the school[17]. The widespread genotype 3.6.1.1.2_CipR.MSM5 was previously reported to be associated with GBMSM in the UK and Australia[10,15]. In our study, male patients were overrepresented (117 male and four female patients), particularly those aged 13 to 68 years (median age: 34 years; interquartile range: 28–42 years), among the cases caused by this genotype (Fig. 3). Furthermore, all but one of the cases infected with this genotype for whom travel information was available reported no travel outside Europe. Our findings indicate that the 164 XDR isolates were found in different transmission networks, with 31 cases in a school setting and eight cases in travelers returning from outside Europe. The male-to-female case ratio suggested that the majority of the other XDR cases may have been associated with GBMSM networks. However, this study was based on routine laboratory-surveillance data, with no record of the sexual orientation of the patient on the notification form accompanying the bacterial isolates sent to the FNRC-ESS (see "Methods") and the patients were not interviewed to determine whether they were GBMSM.

## Analysis of the AMR genes and structures in XDR *S. sonnei* isolates

The combined phenotypic and genomic analyses revealed that all but four of the XDR *S. sonnei* isolates harbored three mutations in the quinolone resistance-determining regions (QRDR) of the *gyrA* and *parC* genes, conferring resistance to CIP (MIC ≥ 1 mg/L) (Figs. 2–4). The other four isolates—from the more ancestral genotypes, 3.6.1_CipR-parent and 3.6.1.1_CipR—harbored only one or two QRDR mutations, but also carried a plasmid-borne *qnr* gene, leading to CIP resistance. Resistance to AZM was conferred by the *mphA* and/or *ermB* genes, whereas resistance to 3GCs was conferred by various extended-spectrum beta-lactamase (ESBL) *bla*$_{CTX-M}$ genes (*bla*$_{CTX-M-3}$, *bla*$_{CTX-M-15}$, *bla*$_{CTX-M-27}$, *bla*$_{CTX-M-55}$, and *bla*$_{CTX-M-134}$) (Figs. 2, 3, 5, and 6).

For better characterization of the mobile (or chromosomally integrated) elements encoding AMR, we also performed long-read sequencing on 16 XDR *S. sonnei* isolates (one to two isolates per XDR genomic cluster). The XDR *S. sonnei* isolates contained one to three plasmids carrying AMR genes (Supplementary Table 3). Plasmid sizes ranged from 8379 to 106,936 kb. A small ~ 8 kb (either non-typable or typed as PTU-E63) plasmid encoding resistance to streptomycin (*strA* and *strB* genes), sulfonamides (*sul2*) and tetracyclines (*tet(A)*) was found in 12 of 16 XDR isolates from all four genotypes. This plasmid was highly similar to pMHMC-012 (GenBank accession no. CP053763) (Supplementary Fig. 1) from a *S. sonnei* isolate collected from a GBMSM patient in Boston, USA in 2017 (ref. [18]).

Most of the other AMR plasmids belonged to IncFII (PTU-FE) and were related to pKSR100 (GenBank accession no. LN624486)—an

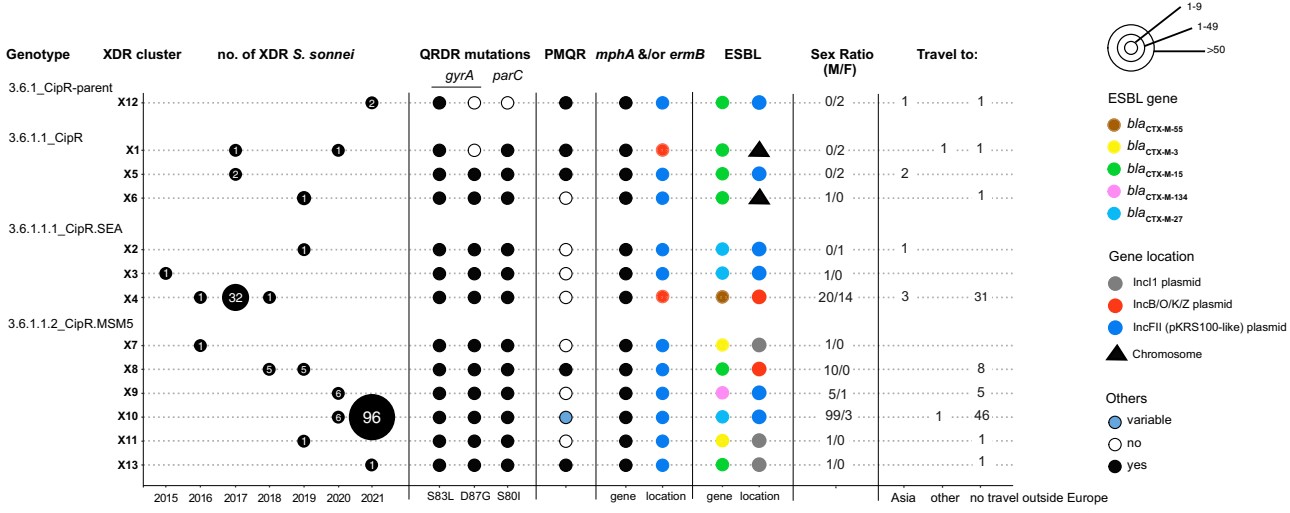

**Fig. 3 | Main characteristics of the 164 XDR *S. sonnei* isolates.** The number of XDR isolates, categorized by genotype and XDR genomic cluster, is given by year of isolation. For each XDR genomic cluster, the mechanisms of resistance to CIP (mutation in the quinolone resistance-determining (QRDR) region of *gyrA* and p*arC*, presence of a plasmid-mediated quinolone (PMQR) resistance gene), AZM (presence of *mph(A)* and/or *erm(B)* resistance genes) or 3GCs (presence of the extended-spectrum beta-lactamase (ESBL) *bla*$_{CTX-M}$ genes indicated in the key) are

indicated. The chromosomal or plasmid location of the AZM and 3GC resistance genes is indicated. For plasmid-borne genes, the type of plasmid is also indicated. Finally, the sex ratio and information about travel (when known) are also indicated for the cases associated with each XDR genomic cluster. The two cases indicated as "Travel to other" are an individual residing in a French overseas territory in the Indian Ocean (cluster X1) and a traveler returning from Central Africa (X10).

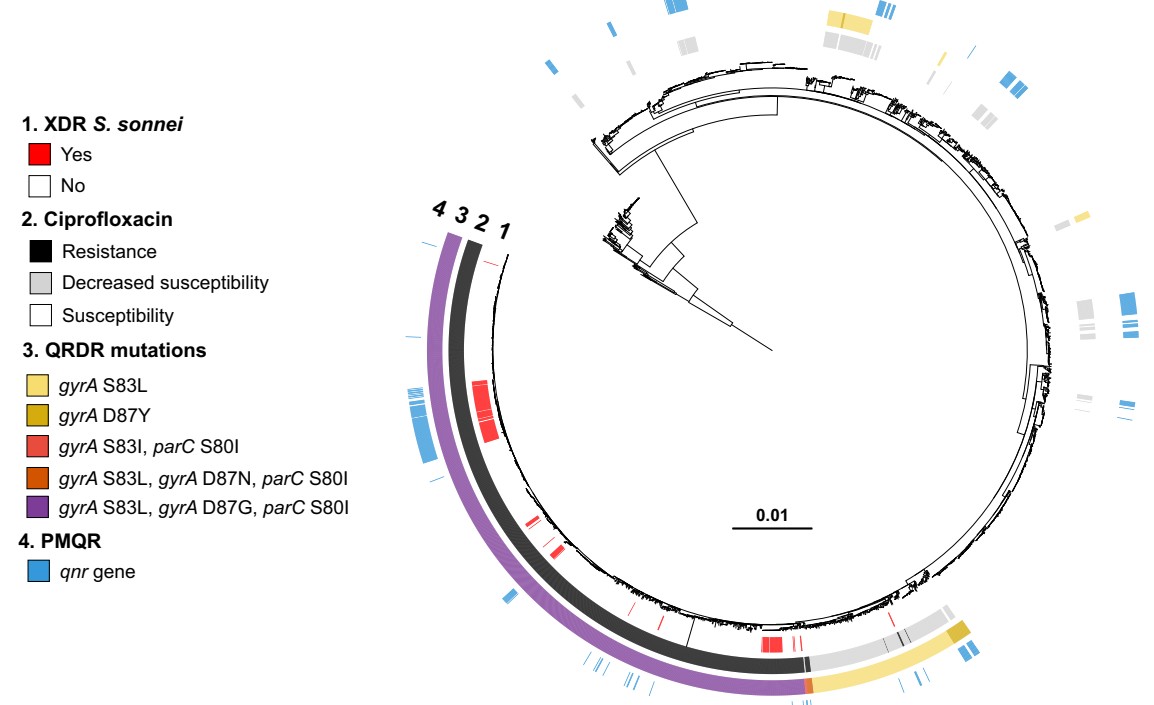

**Fig. 4 | Acquisition of genes encoding resistance to quinolones and fluoroquinolones in our genomic dataset.** Maximum-likelihood phylogeny of 3141 *S. sonnei* genomic sequences as shown in Fig. 2. The rings show the associated information (see key) for each isolate, according to its position in the phylogeny, from the innermost to the outermost, in the following order: (1) the XDR isolates;

(2) antimicrobial susceptibility testing for ciprofloxacin (resistance defined as minimum inhibitory concentration [MIC] ≥ 1 mg/L; susceptibility as MIC ≤ 0.06 mg/L; decreased susceptibility as MIC between 0.12 and 0.5 mg/L); (3) mutations in the quinolone resistance-determining region (QRDR) of *gyrA* and *parC*; (4) presence of plasmid-mediated quinolone resistance (PMQR) genes of the *qnr* family.

epidemic MDR plasmid encoding resistance to AZM with a low fitness cost for its bacterial host − previously described in *S. flexneri* 3a, 2a, and *S. sonnei* sublineages associated with GBMSM networks from Europe, North America and Australasia[10,15,19] (Fig. 3, Supplementary Table 3). A second, less common, plasmid encoding resistance to AZM

was typed as IncB/O/K/Z; plasmids of this type were previously observed in XDR *S. sonnei* isolates in Southeast Asia[6,7] (Fig. 3, Supplementary Table 3).

The ESBL *bla*$_{CTX-M}$ genes were located on multiple plasmids of different types (IncFII, including the pKSR100-like plasmid encoding

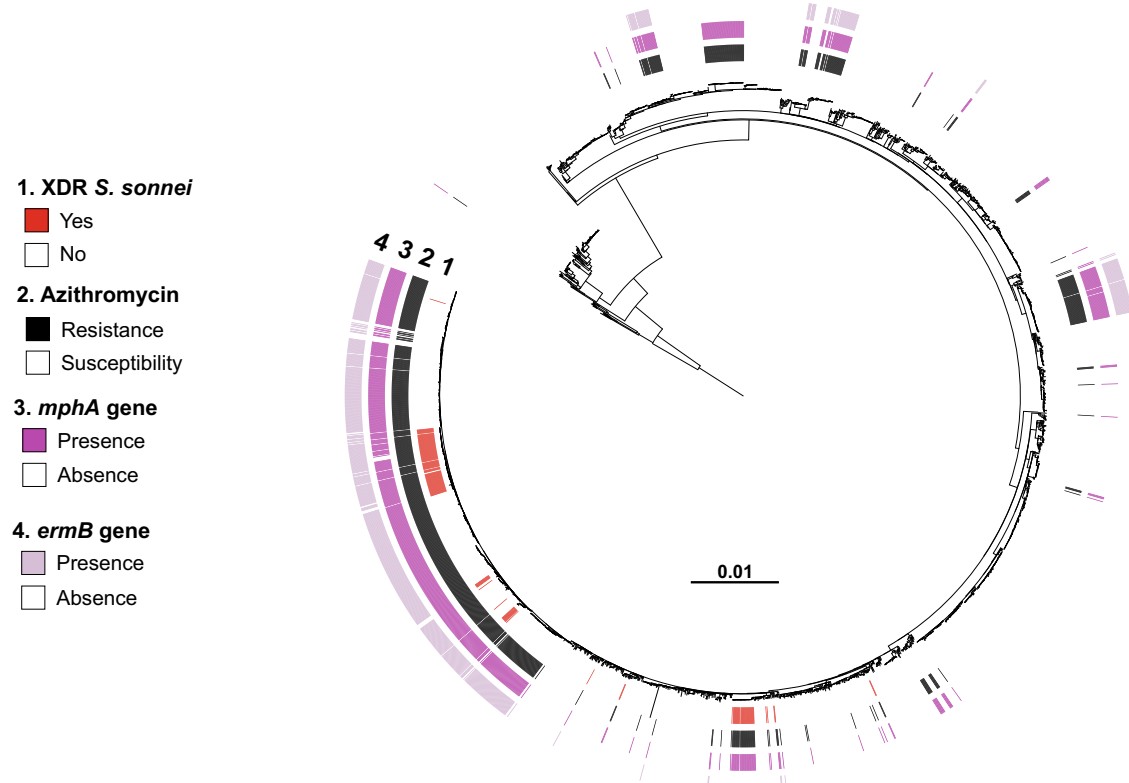

**Fig. 5 | Acquisition of genes encoding resistance to azithromycin in our genomic dataset.** Maximum-likelihood phylogeny of 3141 *S. sonnei* genomic sequences as shown in Fig. 2. The rings show the associated information (see key) for each isolate, according to its position in the phylogeny, from the innermost to the outermost, in the following order: (1) the XDR isolates; (2) antimicrobial susceptibility testing for azithromycin (resistance defined as minimum inhibitory concentration [MIC] ≥ 32 mg/L; susceptibility as MIC ≤ 16 mg/L); (3) presence of the *mph(A)* gene; (4) presence of the *erm(B)* gene.

resistance to AZM, IncB/O/K/Z, IncI1) or even integrated into the bacterial chromosome (Supplementary Table 3). Highly similar ESBL plasmids (98–99% nucleotide identity) were found in different XDR clusters (Supplementary Figs. 2–5). For example, an IncI1 (PTU-I1) plasmid carrying $bla_{CTX-M-3}$ was identified in clusters X7 and X11 (same genotype, 3.6.1.1.2_CipR.MSM5) and an IncFII (PTU-FE) plasmid carrying $bla_{CTX-M-27}$ was identified in clusters X2, X3 and X10 (two different genotypes, 3.6.1.1.1_CipR.SEA and 3.6.1.1.2_CipR.MSM5) (Fig. 3, Supplementary Table 2, and Supplementary Figs. 2 and 3). Some of our ESBL plasmids were also similar to certain plasmids described in previous studies. Hence, the $bla_{CTX-M-27}$-carrying p202008564-6 plasmid (cluster X10, genotype 3.6.1.1.2_CipR.MSM5) displayed 99.6% nucleotide identity to p893916 (GenBank accession no. NZ_MW396858), from a *S. sonnei* isolate collected in London, UK, in 2020 (Supplementary Fig. 2)[20]. The second form of the ESBL plasmid found in cluster X10 isolates—and represented by p202008118-4—probably resulted from an insertion sequence (IS)-driven deletion or acquisition of the azithromycin-resistance gene, *erm(B)* (Supplementary Fig. 2). The $bla_{CTX-M-134}$-carrying p202000562-4 plasmid (cluster X9, genotype 3.6.1.1.2_CipR.MSM5) displayed 90.3% nucleotide identity to p3123885 (GenBank accession no. CP049164), from a *S. sonnei* isolate acquired in Israel in 2019 (Supplementary Fig. 4)[9]. Finally, the $bla_{CTX-M-3}$-carrying p201908234-4 plasmid (cluster X11, genotype 3.6.1.1.2_CipR.MSM5) displayed 99.7% nucleotide identity to p711-69 (GenBank accession no. CP049176), from a *S. sonnei* isolate acquired in Turkey in 2019 (Supplementary Fig. 3)[9]. In two *S. sonnei* isolates, 201701093 (cluster X1, genotype 3.6.1.1_CipR) and 201908033 (cluster X6, also 3.6.1.1_CipR), the $bla_{CTX-M-15}$ ESBL gene was not located on a plasmid but on the bacterial chromosome (Fig. 3, Supplementary Table 3). In isolate 201908033, the ESBL gene was part of a ∼ 42 kb genomic island described in Supplementary Fig. 6, whereas in isolate 201701093, the

ESBL gene was integrated into a ∼ 57 kb prophage sequence absent from other XDR genomes (Supplementary Fig. 7).

In the XDR genomic cluster X10, two isolates, 202110142 (accession no. ERR9940949) and 202111148 (ERR9941000) were phenotypically similar to other X10 isolates, except that they were susceptible to 3GCs (and were therefore not classified as XDR). However, they had the same antimicrobial drug resistance gene content (including $bla_{CTX-M-27}$) as the other X10 XDR isolates. A careful inspection of the assemblies identified an IS (IS26) in both isolates, integrated at the same position of the $bla_{CTX-M-27}$ promotor and probably leading to a non-functional ESBL gene. Together with the emergence of novel AMR genes, the presence of non-functional AMR genes with unaltered open reading frames (ORFs) constitutes a limitation of AMR prediction based exclusively on genomic data.

Our genomic analysis showed that this XDR phenotype emerged through a complex phenomenon of convergent evolution involving many AMR genes and structures. Studies of a larger number of XDR isolates by long-read sequencing might have revealed even greater diversity. Globally, the diversity of mobile and chromosomal AMR elements involved in XDR *S. sonnei* is undoubtedly greater. For example, $bla_{CTX-M-14}$, an ESBL gene not encountered in our study, was identified in seven XDR *S. sonnei* isolates from Australia, UK, Vietnam and the US, between 2015 and 2019 (ref. [15]).

One key element in the generation of the XDR phenotype was the acquisition of resistance to CIP, through fixed chromosomal mutations, in a successful sublineage (3.6.1) in South Asia at some time around 2007 (refs. [6,15]). This sublineage subsequently and successively acquired resistance to AZM and 3GCs. These resistances were acquired independently, on multiple occasions (probably depending on the antibiotic selective pressure exerted in the different *S. sonnei* transmission networks and geographic areas), through the horizontal

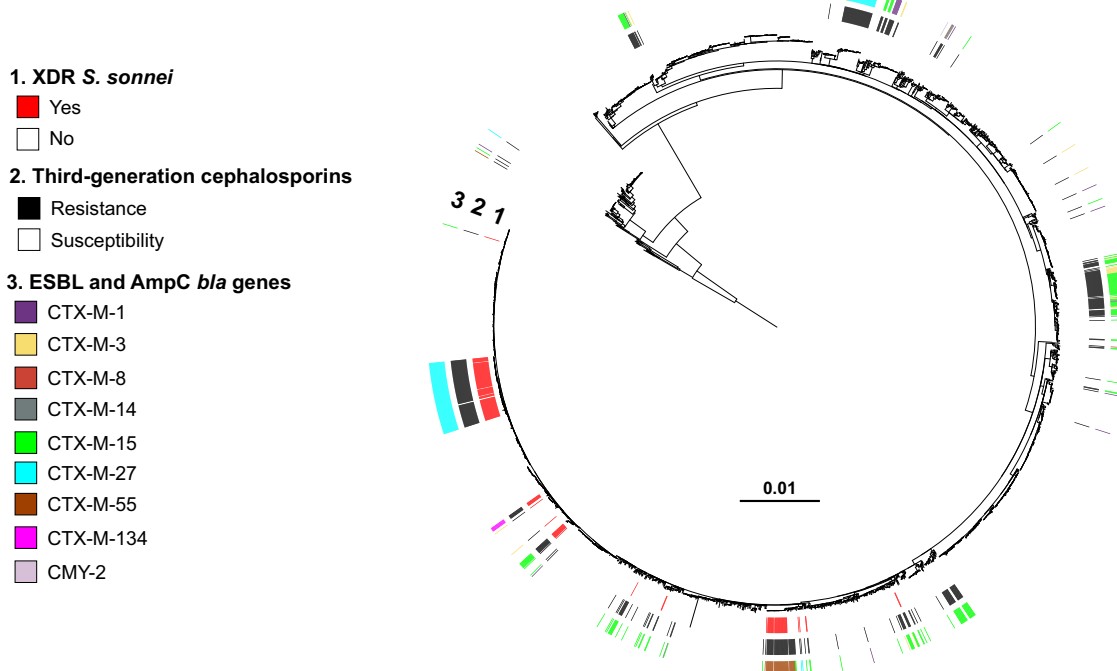

**Fig. 6 | Acquisition of selected genes encoding resistance to third-generation cephalosporins in our genomic dataset.** Maximum-likelihood phylogeny of 3141 *S. sonnei* genomic sequences as shown in Fig. 2. The rings show the associated information (see key) for each isolate, according to its position in the phylogeny, from the innermost to the outermost in the following order: (1) the XDR isolates; (2) antimicrobial susceptibility testing for third-generation cephalosporins (resistance defined as minimum inhibitory concentration [MIC] of ceftriaxone ≥ 4 mg/L; susceptibility as MIC ≤ 1 mg/L); (3) presence of *bla* genes encoding extended-spectrum beta-lactamases (ESBLs) or cephamycinases.

transfer of various plasmids and AMR genes. In the XDR *S. sonnei* isolates collected in France, IncFII pKSR100-like plasmids were frequently the vehicles of resistance to AZM, particularly in GBMSM-associated *S. sonnei* genotypes. Through their acquisition of an additional ESBL gene, these IncFII plasmids were also implicated in the resistance of most of our XDR isolates (clusters X2, X3, X5, X9, X10, and X12), as the vehicles of resistance to C3Gs, ultimately leading to the XDR phenotype. Interestingly, the virulence plasmid (pINV)—a ~210 kb low-copy number non-conjugative plasmid required for the invasion of host epithelial cells by *S. sonnei*—also belongs to the IncFII incompatibility group ([F27: A−: B−] replicon for pINV and [F35: A−: B−] replicon for pKSR100 according to pMLST 2.0 https://cge.food.dtu.dk/services/pMLST/)[21–23]. The coexistence of these two plasmids therefore suggests the existence of two types of selection pressure, mediated by type III secretion system (T3SS) expression[24] and lipolysaccharide biosynthesis[24] for the maintenance of pINV, and antibiotics for the maintenance of pKSR100. The great diversity of ESBL genes ($bla_{CTX-M-3}$, $bla_{CTX-M-15}$, $bla_{CTX-M-27}$, $bla_{CTX-M-55}$, and $bla_{CTX-M-134}$) located on different plasmids (IncFII, IncI1, IncB/O/K/Z) or even integrated into the bacterial chromosome suggests that 3GC agents have recently exerted strong selective pressure on *S. sonnei*. For example, 3GCs are now recommended in place of AZM for the treatment of gonorrhea—a common sexually transmitted infection (STI) in GBMSM—in France and several other European countries[25].

### An emerging XDR *S. sonnei* strain
Our genomic analysis revealed that the 164 French XDR *S. sonnei* isolates were genetically diverse, but with one predominant cluster (X10) accounting for 62.2% (102/164) of XDR isolates. The genomic comparison of these 164 isolates collected in France with 67 previously published international XDR (or inferred to be XDR) *S. sonnei* isolates (Supplementary Table 4) revealed that 32.8% (22/67) of these international isolates were similar to the French XDR isolates from three clusters (X4, X9, and X10) (Supplementary Fig. 8). Eleven international isolates carrying the ESBL $bla_{CTX-M-55}$ gene were identified in our X4 cluster. As expected for this cluster, which was associated with Southeast Asia, five isolates were collected in Vietnam in 2016 (01_0932, 01_0953, 01_0975, 01_1077, and 02_2142), three in Australia in 2016–2017 (AUSMDU00005736, AUSMDU00010002, and AUSMDU00006624; the two first being associated with travel to Southeast Asia), two were obtained in the UK in 2015–2016 (187596 and 261400, with no travel information); and one was obtained in the USA in 2017 (PNUSAE008493, with no travel information). Two international isolates carrying the ESBL $bla_{CTX-M-134}$ gene were found to be similar to our X9 isolates. These two isolates were identified in 2019 in Switzerland (isolate 3123885, from a patient reporting travel to Israel) and in the UK (845732, with no travel information). Our largest XDR cluster, X10—belonging to genotype 3.6.1.1.2_CipR.MSM5 and containing the ESBL $bla_{CTX-M-27}$ gene on a pKSR100-like plasmid—corresponds to the epidemic strain found in GBMSM in the UK and eight other European countries, with a total of 208 isolates obtained as of February 23, 2022 (ref. [5,26]). At the time, France was the country with the largest number of isolates ($n = 106$) and the earliest isolate, dating from September 2020. All 73 XDR *S. sonnei* isolates obtained at the FNRC-ESS from January 1, 2022 to August 31, 2022 (a total of 336 *S. sonnei* isolates were received over this time period) belong to genotype 3.6.1.1.2_CipR.MSM5 and contain the ESBL $bla_{CTX-M-27}$ gene, suggesting that the epidemic X10 strain has continued its intensive circulation in France for a third year. This European epidemic strain was recently identified in the US and Australia[25]. A descriptive epidemiological study of the UK outbreak (72 cases) revealed that 21 of the 27 cases interviewed (78%) were HIV-negative GBMSM and users of HIV pre-exposure prophylaxis (PrEP)[26]. These individuals also reported having high-risk sex in England or Europe during the incubation period. Due to the few therapeutic options left, the recommended oral antimicrobial therapy, when required, was either pivmecillinam or

fosfomycin (chloramphenicol not being readily accessible in the UK)[26]. For hospitalized or severe cases, carbapenems (colistin in case of allergy to beta-lactams) were used for three to five days, followed by an oral step-down treatment[26]. Further studies are required for a more detailed description of the clinical characteristics of these XDR *S. sonnei* infections (severity, existence of asymptomatic carriage, co-morbid conditions, including co-infections with other pathogens, such as monkeypox virus, etc.) and to confirm the efficacy of this anti-microbial treatment strategy. Post-exposure prophylaxis with doxycycline (Doxy-PEP) for the prevention of STIs in GBMSM is now the subject of heated debate[27,28]. A careful evaluation of the balance between the public health risks and benefits of this PEP will need to take the recent emergence of XDR *S. sonnei* into account, as high rates of resistance to tetracyclines (mediated by the two most common genes encoding tetracycline-specific efflux pumps, *tet(A)* and *tet(B)*) (Supplementary Tables 1–3) might provide these bacteria with a selective advantage, favoring their spread.

In conclusion, an effective national laboratory-based surveillance of *Shigella* infections, including antimicrobial susceptibility data, is therefore essential for informed decision-making and appropriate public health action to tackle the spread of these XDR *S. sonnei* strains currently circulating in different transmission networks. Genomic surveillance based on this new genotyping scheme with a common nomenclature will facilitate the detection and global tracking of these XDR *S. sonnei* strains of concern.

## Methods

### Ethics statement
This study was based exclusively on bacterial isolates and associated metadata collected under the mandate for laboratory-based surveillance awarded by the French Ministry of Health to the National Reference Center for *Escherichia coli*, *Shigella* and *Salmonella* (NRC-ESS). As a result, neither informed consent nor approval from an ethics committee was required. Data collection and storage by the NRC-ESS was approved by the French National Commission for Data Protection and Liberties ("*Commission Nationale Informatique et Libertés* (CNIL)"; approval number: 1474659).

### Bacterial isolates
The French national surveillance program for *Shigella* infections is based on a voluntary laboratory-based network consisting of approximately 1000 clinical laboratories located in mainland France and its overseas territories in South America (French Guiana), the Caribbean (Martinique, Guadeloupe) and the Indian Ocean (La Réunion, Mayotte), which send about 1000–1200 *Shigella* spp. isolates to the French National Reference Center for *E. coli*, *Shigella* and *Salmonella* (FNRC-ESS) at Institut Pasteur, each year (only 600 in 2020, due to the COVID pandemic). The *Shigella* isolates are sent to the FNRC-ESS with a notification form that includes basic data: patient name, date of birth, sex, postcode, type of sample (stools or other), isolation date, clinical symptoms and their onset, illness or asymptomatic carriage (if illness, types of symptoms), information about international travel (if yes, date and country), sporadic or outbreak isolates (if clustered, hospital, school, household, nursery, etc.).

We studied all 7121 *S. sonnei* isolates (one per patient) received by the FNRC-ESS between 2005 and 2021, in the framework of the French national surveillance program for *Shigella* infections. It has been estimated that this surveillance system detects 50–60% of laboratory-confirmed *Shigella* infections in France[29]. The 7121 *S. sonnei* human isolates studied consisted of 6700 (94.1%) from mainland France and 421 (5.9%) from French overseas territories. From January 2005 to September 2021 (when phenotypic typing was definitively replaced by genomic surveillance), all these isolates were thoroughly characterized with a panel of biochemical tests and serotyped with slide agglutination assays according to standard protocols, as previously described[30].

### Antimicrobial drug susceptibility testing
Antimicrobial drug susceptibility testing was performed on all *S. sonnei* isolates, at the time of reception. Isolates were first tested with the disk diffusion (DD) method on Mueller-Hinton agar (Bio-Rad, Marnes-la-Coquette, France) according to the 2005–2015 guidelines of the anti-biogram committee of the French Society for Microbiology (CA-SFM), in accordance with the recommendations of the European Committee on Antimicrobial Susceptibility Testing (EUCAST) (https://www.sfm-microbiologie.org/casfm/). The following disks (Bio-Rad, Marnes-La-Coquette, France) were used for the DD method: amoxicillin (AMX, 10 μg) or ampicillin (AMP, 10 μg), cefotaxime (CTX, 5 μg), ceftazidime (CAZ, 30 μg before 2015, 10 μg from 2015), ertapenem (ETP, 10 μg), chloramphenicol (CHL, 30 μg), sulfonamides (SMX, 200 μg), trimethoprim (TMP, 5 μg), trimethoprim-sulfamethoxazole (SXT, 1.25 μg/23.75 μg), streptomycin (STR, 10 μg), amikacin (AKN, 30 μg), gentamicin (GEN, 10 μg), tetracycline (TET, 30 μg), nalidixic acid (NAL, 30 μg), ofloxacin (OFX, 5 μg) or pefloxacin (PEF, 5 μg), ciprofloxacin (CIP, 5 μg), and azithromycin (AZM, 15 μg). Susceptibility to AZM was tested systematically only from April 2014. Resistance to 3GCs was defined as resistance to ceftazidime, cefotaxime, or ceftriaxone. For isolates resistant to NAL, CIP, AZM, or 3GCs, we confirmed the DD data by determining the minimum inhibitory concentrations (MICs) of these drugs with Etest strips (AB Biodisk, Solna, Sweden; bioMérieux, Marcy L'Etoile, France). The Clinical and Laboratory Standards Institute (CLSI) criteria were then used for the final interpretation[31]. As a means of distinguishing *Shigella* isolates susceptible to ciprofloxacin (minimum inhibitory concentration [MIC] ≤ 0.25 mg/L) that are wild-type (WT) from those that are non-WT, we defined two categories based on the epidemiological cutoffs used by the CLSI for *Salmonella* spp.: decreased susceptibility to ciprofloxacin (MIC between 0.12 and 0.5 mg/L) and true susceptibility to ciprofloxacin (MIC ≤ 0.06 mg/L)[31].

### Whole-genome sequencing
In total, 3109 *S. sonnei* isolates from the 7121 collected between 2005 and 2021 were sequenced, including all 164 XDR isolates (Supplementary Tables 1 and 2). Between 2017 (the year in which genomic surveillance began in France) and 2021, 2618 clinical isolates were received and sequenced at the FNRC-ESS. We included in this study the 2558 genomes (97.7%) that passed the EnteroBase quality control criteria (https://enterobase.warwick.ac.uk/species/index/ecoli). We also sequenced a selection of 551/4503 (12.2%) of the 2005–2016 isolates. This selection contained, in particular, 96.7% (89/92) of all isolates resistant to 3GCs, 56.3% (294/522) of all isolates resistant to CIP, and 96.7% (116/120) of the isolates resistant to AZM detected from April 2014 to December 2016.

Total DNA was extracted with the MagNA Pure DNA isolation kit (Roche Molecular Systems, Indianapolis, IN, USA), in accordance with the manufacturer's recommendations. Whole-genome sequencing was performed as part of routine procedures at the FNRC-ESS, and at the Mutualized Platform for Microbiology (P2M) at Institut Pasteur, Paris. The libraries were prepared with the Nextera XT kit (Illumina, San Diego, CA, USA) and sequencing was performed with the NextSeq 500 system (Illumina) generating 150 bp paired-end reads. All reads were filtered with FqCleanER version 21.06 (https://gitlab.pasteur.fr/GIPhy/fqCleanER) with options -q 15 -l 50 to eliminate adaptor sequences and discard low-quality reads with phred scores below 15 and a length of less than 50 bp[32].

Assemblies were generated with SPAdes version 3.9.0 (ref. [33]) through EnteroBase.

### Genotyping
All the genomes studied were genotyped with the hierarchical SNV-based genotyping scheme for *S. sonnei* described by Hawkey et al.[15] and implemented in Mykrobe software version 0.9.0 (https://github.com/katholt/sonneityping)[34].

## Phylogenomic analysis

We included in the phylogenomic analysis 3140 *S. sonnei* genomic sequences from isolates and historical strains of the FNRC-ESS (Supplementary Data 1) originating from 3109 *S. sonnei* isolates collected between 2005 and 2021 and 31 isolated before 2005, to enrich the dataset with rare lineages of *S. sonnei* (L1, L2, and L4)[6,8,10,16].

The paired-end reads were mapped onto the reference genome of *S. sonnei* 53G (GenBank accession numbers NC_016822)[16] with Snippy version 4.6.0/BWA-MEM version 0.7.17 (https://github.com/tseemann/snippy). SNVs were called with Snippy version 4.6.0/Freebayes version 1.3.2 (https://github.com/tseemann/snippy) under the following constraints: mapping quality of 60, a minimum base quality of 13, a minimum read coverage of 4, and a 75% read concordance at a locus for a variant to be reported. An alignment of core genome SNVs was produced in Snippy version 4.6.0 for phylogenetic inference.

Repetitive regions (i.e., insertion sequences, tRNAs) in the alignment were masked (https://doi.org/10.26180/5f1a443b19b2f)[15]. Putative recombinogenic regions were detected and masked with Gubbins version 3.2.0 (ref. [35]) (default settings, except -f 32). A maximum likelihood (ML) phylogenetic tree was built from an alignment of 59,295 chromosomal SNVs, with RAxML version 8.2.12, under the GTR model, with 200 bootstrap values[36]. The final tree was rooted on the *S. flexneri* 2a strain 2457T genome (GenBank accession no. AE014073) and visualized with iTOL version 6 (https://itol.embl.de)[37].

We also used the same phylogenetic approach to compare our 164 XDR *S. sonnei* isolates with 69 previously published XDR (or inferred to be XDR) *S. sonnei* isolates (Supplementary Data 1), but with Gubbins used with default settings (in particular -f 25). The final tree, based on 2298 chromosomal SNVs, was rooted on the reference *S. sonnei* genome 53G (genotype 2.8.2). Two of these international isolates (Vietnamese isolates 02_1181 and 01_1008) were excluded from the analysis due to a proportion of "*N*" > 25%.

## Resistance gene analysis

The presence and type of acquired antibiotic resistance genes (ARGs) were determined with ResFinder version 4.0.1 (https://cge.cbs.dtu.dk/services/ResFinder/)[38], Sonneityping/Mykrobe version 0.9.0 (https://github.com/katholt/sonneityping)[15,34] on SPAdes assemblies. The presence of mutations in genes encoding resistance to quinolones (*gyrA*, *parC*) was also investigated by analyzing the sequences assembled de novo with BLAST version 2.2.26.

## Plasmid sequencing

Sixteen XDR *S. sonnei* isolates (one to two per XDR cluster) were selected (Supplementary Table 3) and sequenced with a Nanopore MinION sequencer (Oxford Nanopore Technologies). Genomic DNA was prepared as follows: the isolates were cultured overnight at 37 °C in alkaline nutrient agar (20 g casein meat peptone E2 from Organotechnie; 5 g sodium chloride from Sigma; 15 g Bacto agar from Difco; distilled water to 1 L; adjusted to pH 8.4; autoclaved at 121 °C for 15 min). A few isolated colonies from the overnight culture were used to inoculate 20 mL of brain-heart infusion (BHI) broth, and were cultured until a final $OD_{600}$ of 0.8 was reached at 37 °C with shaking (200 rpm— Thermo Fisher Scientific MaxQ 6800). The bacterial cells were harvested by centrifugation and DNA was extracted with one of the two following methods. The first method corresponded to the protocol described by von Mentzer et al.[39], except that MaXtract High-Density columns (Qiagen) were used (instead of phase-lock tubes) and the DNA was resuspended in molecular biology-grade water (instead of 10 mM Tris pH 8.0). In the second method, we used Genomic-tip 100/G columns (Qiagen) according to the manufacturer's protocol. The library was prepared according to the instructions of the "Native barcoding genomic DNA (with EXP-NBD104, EXP-NBD114, and SQK-LSK109)" procedure provided by Oxford Nanopore Technology. Sequencing was then performed on a MinION Mk1C apparatus (Oxford Nanopore

Technologies). The genomic sequences of the isolates were assembled from long and short reads, with a hybrid approach and UniCycler version 0.4.8 (ref. [40]). A polishing step was performed with Pilon version 1.23 (ref. [41]) to generate a high-quality sequence composed of chromosomal and plasmid sequences. The plasmids were then annotated with Prokka version 1.14.5 (https://github.com/tseemann/prokka)[42] and corrected manually. Plasmids were aligned and visualized with BRIG version 0.95 (http://sourceforge.net/projects/brig)[43].

## Plasmid typing

The plasmids were typed with PlasmidFinder version 2.1.1. (https://cge.cbs.dtu.dk/services/PlasmidFinder/)[44], pMLST version 1.2 (https://cge.cbs.dtu.dk/services/pMLST/)[44], and COPLA version 1.0 (https://castillo.dicom.unican.es/copla/)[45] on SPAdes assemblies.

## Statistical analysis

Chi-squared tests for trends were used to analyze the proportion of bacterial strains resistant to antimicrobial drugs by year.

## Data collection

The data were entered into an Excel (Microsoft) version 15.41 spreadsheet.

## Reporting summary

Further information on research design is available in the Nature Portfolio Reporting Summary linked to this article.

# Data availability

The publicly available sequences used in this study are available in GenBank under accession numbers CP053763, NC_016822, AE014073, NZ_MW396858, LN624486, CP049176, CP049164, CP049174, CP049186, CP053751. Short-read sequence data generated in this study were submitted to EnteroBase (https://enterobase.warwick.ac.uk/) and to the European Nucleotide Archive (ENA, https://www.ebi.ac.uk/ena/) under study number PRJEB44801. Whole-genome assemblies have been deposited in FigShare (https://doi.org/10.6084/m9.figshare.21594033.v1). All the accession numbers of the short-read sequences produced and used in this study are listed in Supplementary Table 2 and Supplementary Data 1. The plasmid sequences obtained were deposited in GenBank (https://www.ncbi.nlm.nih.gov/genbank/) under accession numbers OP038267-OP038301 and OP038303 (Supplementary Table 3).

# Code availability

No custom computer code or custom algorithm was used in this study.

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

## Acknowledgements

This study was supported by the *Institut Pasteur* (F.-X.W. and A.F.), *Santé publique France* (F.-X.W.), and the French government's *Investissement d'Avenir* program, *Laboratoire d'Excellence* 'Integrative Biology of Emerging Infectious Diseases' (grant number ANR-10-LABX-62-IBEID) (F.-X.W.). We thank C. Parsot for his expert advice on the virulence plasmid of *Shigella* spp. We also thank all the corresponding laboratories of the

French National Reference Center for *Escherichia coli, Shigella,* and *Salmonella*. The funders had no role in study design, data collection and analysis, decision to publish, or preparation of the manuscript.

## Author contributions

S.L. and F.-X.W. conceptualized and designed the study. S.L., E.N., I.Y., and F.-X.W. did the genomic analyses. S.L., E.N., I.Y., L.F., and F.-X.W. contributed to data interpretation and visualization. C.R., I.C., M.L.C., and E.N. performed the laboratory experiments. S.L. and S.F. were involved in sample collection and metadata curation. S. F. and A.F. performed statistical analyses. F.-X.W. supervised the project. F.-X.W. and A.F. were responsible for funding acquisition. F.-X.W. drafted the article. S.L., E.N., S.F., L.F., I.Y., L.F., M.P.G., and A.F. critically reviewed the draft. All the authors read and approved the final manuscript. S.L., E.N., and F.-X.W. accessed and verified the underlying data.

## Competing interests

The authors declare no competing interests.
