## [Peer Review File · Nature Communications]

Rapid emergence of extensively drug-resistant *Shigella sonnei* in FranceEditorial Note: This manuscript has been previously reviewed at another journal that is not operating a transparent peer review scheme. This document only contains reviewer comments and rebuttal letters for versions considered at *Nature Communications*.

REVIEWER COMMENTS

Reviewer #4 (Remarks to the Author):

The revised "Recent and rapid emergence of extensively drug-resistant *Shigella sonnei* in France" describes the integration of Whole Genome Sequencing into the public health framework to describe the emergence and examination of drug resistant *Shigella sonnei* in France. The authors have addressed a number of the previous comments adequately; however, there are a number that are not well addressed and the most important are highlighted below.

1) The description of the selection of the *E. coli* 045 isolate as the root for the phylogenetic analysis is not well reasoned. I agree with the authors definition of an outgroup isolate, but there is still no rationale as to WHY this isolate was selected? Additionally, the *S. sonnei* are as different from any other serogroup/species of *Shigella* as they are from this seemingly random *E. coli* isolate. One could also make the case that the use of another *Shigella* serogroup/species isolate would add gravitas to the analysis. The cited articles identified in this rebuttal all support this point. The suggestion that the "novel" (as defined by the authors) Hawkeye tool, does not provide the additional support that the authors think. These analyses should be completed using a non-*sonnei* *Shigella* isolate as the outgroup. Additionally, the removal of recombination from the analysis seems counterintuitive, in that the authors are examining the genomes for regions of mobility that would be associated with development of resistance, yet have removed all of these regions from the analysis. This severely limits the utility, accuracy and conclusions that can be drawn from this study.

2) The authors have misunderstood the point about plasmid loss that was previously raised. Traditionally any genomic study of *S. sonnei* has lacked the examination of the virulence plasmids, which are highly unstable. There is a false equivalency to try and compare to other enterics, even to other *Shigella*, as *E. coli* 045 isolates, *S. flexneri* nor *S. boydii* have the same labile plasmid issues. This is not addressed in this manuscript and if this has any impact on the plasmids that are harboring the AMR genes and their stability (most investigators assume that the *S. sonnei* plasmid is identical, but there is little data to support this conjecture, but these additional plasmid replicons may play a significant role in the acquisition and stability of the AMR containing plasmids.

3) The authors have indicated there is another paper with greater geographic distribution over greater time frames as a preprint – it may be worthwhile to highlight if there were any significant differences with the data from France, that was not observed in other countries. What makes this study stand alone and novel? This is unclear and this study is possibly not novel?

4) Additionally, the authors should take it under advisement that the responding to a reviewers comment with "We previously made clear..." would seem to suggest that the authors believe there is no comment to be addressed, or that they have already addressed the comment; however, if someone is taking the time to review the manuscript and ask the question the authors may want to question if it is really as clear as they think it is.

REVIEWER COMMENTS

Reviewer #4 (Remarks to the Author):

The revised “Recent and rapid emergence of extensively drug-resistant *Shigella sonnei* in France” describes the integration of Whole Genome Sequencing into the public health framework to describe the emergence and examination of drug resistant *Shigella sonnei* in France. The authors have addressed a number of the previous comments adequately; however, there are a number that are not well addressed and the most important are highlighted below.

1) The description of the selection of the *E. coli* O45 isolate as the root for the phylogenetic analysis is not well reasoned. I agree with the authors definition of an outgroup isolate, but there is still no rationale as to WHY this isolate was selected? Additionally, the *S. sonnei* are as different from any other serogroup/species of *Shigella* as they are from this seemingly random *E. coli* isolate. One could also make the case that the use of another *Shigella* serogroup/species isolate would add gravitas to the analysis. The cited articles identified in this rebuttal all support this point. The suggestion that the “novel” (as defined by the authors) Hawkeye tool, does not provide the additional support that the authors think. These analyses should be completed using a non-*sonnei* *Shigella* isolate as the outgroup. Additionally, the removal of recombination from the analysis seems counterintuitive, in that the authors are examining the genomes for regions of mobility that would be associated with development of resistance, yet have removed all of these regions from the analysis. This severely limits the utility, accuracy and conclusions that can be drawn from this study.

Authors: We have redone the phylogenetic analysis with *Shigella flexneri* 2a strain 2457T (GenBank accession number AE014073) as the outgroup. We obtained a clustering (i.e., 13 clusters of XDR isolates in the *S. sonnei* clade 3.6.1) similar to that obtained in the analysis in which *E. coli* O45 was used as the outgroup.

It is standard practice in the field to investigate vertical evolution and horizontal gene transfer (HGT) separately (Holt, K.H. et al. Nat Genet 2012; Njamkepo, E. et al. Nat Microbiol 2016; Chung The, H. et al. Nat Commun 2019; Hawkey, J. et al. Nat Commun 2021; Bengtsson, R.J. et al. Nat Microbiol 2022). As a means of avoiding the conflation of these two different evolutionary mechanisms, we used a recombination-filtered tree, which provides an accurate view of the evolutionary relationships between *Shigella* strains (vertical evolution). We then analyzed HGT, for mobile AMR determinants in particular.

2) The authors have misunderstood the point about plasmid loss that was previously raised. Traditionally any genomic study of *S. sonnei* has lacked the examination of the virulence plasmids, which are highly unstable. There is a false equivalency to try and compare to other enterics, even to other *Shigella*, as *E. coli* O45 isolates, *S. flexneri* nor *S. boydii* have the same labile plasmid issues. This is not addressed in this manuscript and if this has any impact on the plasmids that are harboring the AMR genes and their stability (most investigators assume that the *S. sonnei* plasmid is identical, but there is little data to support this conjecture, but these additional plasmid replicons may play a significant role in the acquisition and stability of the AMR containing plasmids.

Authors: We apologize for not having understood that the reviewer was specifically referring to virulence plasmid (pINV) loss. Since the pioneering work of Philippe Sansonetti (Sansonetti et al. Infect Immun. 1981) 42 years ago, the presence of pINV has been known to be essential for virulence in human cells. This virulence trait was subsequently shown to be mediated by a type III secretion system (T3SS) encoded by pINV (Maurelli et al. Infect Immun 1985). A loss of pINV is frequently observed in the laboratory and leads to avirulent *S. sonnei* (i.e., which can no longer invade epithelial cells, Martyn et al. J Bacteriol. 2022;204: e0051921). We have discussed the possible interactions of pINV and AMR plasmids with Claude Parsot, who has studied pINV in Sansonetti's laboratory for the last 30 years. According to him, it is very difficult to study pINV due to the rapid loss of this plasmid after isolation on a solid culture medium (without the use of Congo red dye, which can be used to select pINV⁺ colonies, which are red in the presence of this stain) and the presence of several hundreds of insertion sequences (ISs) within pINV, precluding the use of short-read sequencing to study this large (over 200 kb) plasmid. Our study was not designed to investigate interactions between pINV and AMR plasmids. However, we now indicate in the revised manuscript (lines 220 to 227) that most French XDR *S. sonnei* unexpectedly contained two plasmids (pINV and pKRS100) from the same incompatibility group (IncFII), suggesting a double selection pressure. The first selection pressure was exerted by the need to express the T3SS and lipopolysaccharide to ensure virulence (maintenance of pINV) and the second selection pressure was exerted by antibiotics (maintenance of pKRS100).

3) The authors have indicated there is another paper with greater geographic distribution over greater time frames as a preprint – it may be worthwhile to highlight if there were any significant differences with the data from France, that was not observed in other countries. What makes this study stand alone and novel? This is unclear and this study is possibly not novel?

Authors: The second paper (actually a preprint released after ours) by Mason et al. describes the international spread (France included) of the XDR epidemic X10 strain in GBMSM networks. This other study is essentially a genomic study (only 14 isolates with antimicrobial susceptibility testing) and does not cover a larger time frame (2016-2022) than our study. We did not claim our study was novel. However, our study provides, for the first time, a precise timeline of the emergence, spread and genomic evolution of XDR *S. sonnei* at national level over a period of 17 years (2005-2021). Furthermore, our study is not restricted to a single cluster of XDR strains in GBMSM networks (as for the X10 epidemic strain in the article by Mason et al.), but instead describes 13 different clusters with various transmission networks.

In this revised MS, we now compare the genomic sequences of the 164 French XDR isolates with those of 67 international XDR isolates (listed in new Supplementary Table 5) reported before the submission of our paper. Most of these international isolates were actually only inferred to be XDR, as antimicrobial susceptibility data are lacking for confirmation of the XDR phenotype. We found that only 33% (22/67) of these international XDR isolates were similar to the French XDR isolates. These isolates belonged to only three (X4, X9 and X10) of our 13 XDR clusters. We have added the following text “*The genomic comparison of these 164 isolates collected in France with 67 previously published international XDR (or inferred to*

be XDR) S. sonnei isolates (Supplementary Table 5) revealed that 32.8% (22/67) of these international isolates were similar to the French XDR isolates from three clusters (X4, X9 and X10) (Supplementary Fig. 8). Eleven international isolates carrying the ESBL bla_{CTX-M-55} gene were identified in our X4 cluster. As expected for this cluster, which was associated with Southeast Asia, five isolates were collected in Vietnam in 2016 (01_0932, 01_0953, 01_0975, 01_1077 and 02_2142), three in Australia in 2016-2017 (AUSMDU00005736, AUSMDU00010002 and AUSMDU00006624; the two first being associated with travel to Southeast Asia), two were obtained in the UK in 2015-2016 (187596 and 261400, with no travel information); and one was obtained in the USA in 2017 (PNUSAE008493, with no travel information). Two international isolates carrying the ESBL bla_{CTX-M-134} gene were found to be similar to our X9 isolates. These two isolates were identified in 2019 in Switzerland (isolate 3123885, from a patient reporting travel to Israel) and in the UK (845732, with no travel information).” in lines 238-251 and new Supplementary Figure 8.

We reiterate that the two studies are truly complementary, showing the timeline for emergence and the diversity of XDR *S. sonnei* isolates in various transmission networks at national level (Lefèvre et al.) and the international epidemic potential of one of these XDR strains through its spread in GBMSM networks (Mason et al.).

4) Additionally, the authors should take it under advisement that the responding to a reviewers comment with “We previously made clear....” would seem to suggest that the authors believe there is no comment to be addressed, or that they have already addressed the comment; however, if someone is taking the time to review the manuscript and ask the question the authors may want to question if it is really as clear as they think it is.

Authors: We apologize for any offense our response may have caused. We did not wish to imply that the reviewer’s comment was irrelevant or not worthy of careful consideration. We were simply trying to indicate what we had already stated in the manuscript, so as to clarify our response.